# Intermittent Lead Exposure Induces Behavioral and Cardiovascular Alterations Associated with Neuroinflammation

**DOI:** 10.3390/cells12050818

**Published:** 2023-03-06

**Authors:** Liana Shvachiy, Ângela Amaro-Leal, Tiago F. Outeiro, Isabel Rocha, Vera Geraldes

**Affiliations:** 1Department of Experimental Neurodegeneration, Center for Biostructural Imaging of Neurodegeneration, University Medical Center Göttingen, 37075 Göttingen, Germany; 2Cardiovascular Centre of the University of Lisbon, 1649-028 Lisbon, Portugal; 3Institute of Physiology, Faculty of Medicine of the University of Lisbon, 1649-028 Lisbon, Portugal; 4Max Planck Institute for Natural Science, 37075 Göttingen, Germany; 5Translational and Clinical Research Institute, Faculty of Medical Sciences, Newcastle University, Framlington Place, Newcastle Upon Tyne NE2 4HH, UK; 6Scientific Employee with an Honorary Contract at Deutsches Zentrum für Neurodegenerative Erkrankungen (DZNE), 37073 Göttingen, Germany

**Keywords:** intermittent lead exposure toxicity, long-term memory impairment, hypertension, baroreflex impairment, neuroinflammation, synaptic dysfunction

## Abstract

The nervous system is the primary target for lead exposure and the developing brain appears to be especially susceptible, namely the hippocampus. The mechanisms of lead neurotoxicity remain unclear, but microgliosis and astrogliosis are potential candidates, leading to an inflammatory cascade and interrupting the pathways involved in hippocampal functions. Moreover, these molecular changes can be impactful as they may contribute to the pathophysiology of behavioral deficits and cardiovascular complications observed in chronic lead exposure. Nevertheless, the health effects and the underlying influence mechanism of intermittent lead exposure in the nervous and cardiovascular systems are still vague. Thus, we used a rat model of intermittent lead exposure to determine the systemic effects of lead and on microglial and astroglial activation in the hippocampal dentate gyrus throughout time. In this study, the intermittent group was exposed to lead from the fetal period until 12 weeks of age, no exposure (tap water) until 20 weeks, and a second exposure from 20 to 28 weeks of age. A control group (without lead exposure) matched in age and sex was used. At 12, 20 and 28 weeks of age, both groups were submitted to a physiological and behavioral evaluation. Behavioral tests were performed for the assessment of anxiety-like behavior and locomotor activity (open-field test), and memory (novel object recognition test). In the physiological evaluation, in an acute experiment, blood pressure, electrocardiogram, and heart and respiratory rates were recorded, and autonomic reflexes were evaluated. The expression of GFAP, Iba-1, NeuN and Synaptophysin in the hippocampal dentate gyrus was assessed. Intermittent lead exposure induced microgliosis and astrogliosis in the hippocampus of rats and changes in behavioral and cardiovascular function. We identified increases in GFAP and Iba1 markers together with presynaptic dysfunction in the hippocampus, concomitant with behavioral changes. This type of exposure produced significant long-term memory dysfunction. Regarding physiological changes, hypertension, tachypnea, baroreceptor reflex impairment and increased chemoreceptor reflex sensitivity were observed. In conclusion, the present study demonstrated the potential of lead intermittent exposure inducing reactive astrogliosis and microgliosis, along with a presynaptic loss that was accompanied by alterations of homeostatic mechanisms. This suggests that chronic neuroinflammation promoted by intermittent lead exposure since fetal period may increase the susceptibility to adverse events in individuals with pre-existing cardiovascular disease and/or in the elderly.

## 1. Introduction

Excessive accumulation of lead in nerve cells leads to neurotoxic effects [1,2,3]. Indeed, the nervous system is the primary target of lead exposure in all age groups, but the brain has been shown to be the structure most sensitive to toxic lead exposure, especially during development [3,4,5,6,7].

Currently, the mechanisms of neurotoxicity of lead are still unclear, but microgliosis and astrogliosis have been considered as possible processes to explain the toxic effects on the nervous system, triggering a neuroinflammatory cascade and disrupting neural pathways involved in hippocampal functions [2,8,9,10].

In particular, various inflammatory cytokines are produced by astroglial cells due to insufficient energy transfer from astrocytes to neurons caused by chronic overexposure to occupational or environmental lead sources [11]. Neuroinflammation is, then, caused by protracted activation of microglia and astrocytes and is associated with infections, autoimmunity, and the development of neurodegenerative diseases such as Alzheimer’s disease and amyotrophic lateral sclerosis [12,13,14,15,16,17,18,19]. Furthermore, in response to lead exposure, microglia and astrocytes can increase the production and release of reactive oxygen species (ROS), such as hydroperoxides, and inhibit antioxidant activity because lead can exchange electrons between the lead group and the sulfhydryl groups in antioxidant enzymes, inducing neuronal damage and apoptosis [10,20,21,22,23,24,25,26,27,28].

Neuroinflammation triggered by astrocytes and microglia has also been directly linked to cardiovascular disease [29,30,31,32,33,34]. Indeed, some studies suggest that microglia play a key role in the neurogenic regulation of hypertension, and astrocyte activation has also been linked to the modulation of the baroreceptor reflex in areas of the central autonomic network such as the hypothalamus [34,35].

In animal models with lead exposure, cognitive deficits related to learning and memory have also been observed, with hippocampal cells being particularly affected by a long-term increased signaling between them [1,2,4,36,37,38]. Along with long-term potentiation, lead appears to impair learning through changes in neuronal and glial cells of the hippocampus, affecting glutamate release, NMDA receptor function and structural plasticity, thereby altering synaptic plasticity [8,39,40,41,42,43].

Lead neurotoxicity may manifest differently depending on the type of exposure, but studies of intermittent lead exposure associated with human and societal behaviors such as globalization, increased migration, and academic or occupational exchange programs are rare. Only a few studies have been performed before to examine the new paradigm of lead exposure, the intermittent environmental lead exposure; Bihaqi was one of the first to study this type of exposure, albeit only from the post-natal period [15]. Additionally, we have previously presented the effects of intermittent lead exposure and compared them to the different paradigms of low-level environmental lead poisoning [44,45,46]. Based on the physiological, autonomic, behavioral and molecular changes induced by different low-level environmental lead exposures from fetus to adulthood, we have shown that the duration of exposure rather than the timing of exposure in life is the main factor for more severe adverse and toxic effects [45,46]. In particular, the effects of an intermittent Pb exposure evaluated in one time-point (at 28 weeks of age) are similar to a permanent exposure; however, they are less pronounced due to the shorter duration of exposure [45]. Nevertheless, we do not know the time-point at which these adverse effects appear, nor whether these effects are due to fetal exposure.

Therefore, in this longitudinal study, we sought to understand the relationship between lead-induced neurodegeneration and the development of the molecular, physiological, and behavioral changes associated with intermittent lead exposure at three timepoints. The complementary roles of astrocytes and microglia in defining brain synaptic function were also explored to clarify the ways in which their cellular crosstalk might contribute to facilitating the neurotoxic effects of lead in the hippocampus.

## 2. Materials and Methods

### 2.1. Animal Model Development

Since lead ingestion is one of the three primary ways that lead enters the body, an animal model of lead exposure was created, as previously reported [47,48]. Wistar rats that were seven days pregnant were separated into Pb-treated and control groups (Charles River Laboratories, Chatillon-sur-Chalaronne, France). The tap drinking water in the Pb-treated group was changed to a 0.2% (*p*/*v*) lead (II) acetate solution dissolved in deionized water (equivalent to 2000 ppm or 2000 mg/L of the solution which is considered a low-level environmental lead exposure in humans) (Acros Organics, New Jersey, NJ, USA).

Rat pups of both groups received the same lead solution: for intermittent (IntPb) exposure to lead until 12 weeks of age, no exposure (tap water) until 20 weeks, and second exposure from 20 to 28 weeks of age (*n* = 14); tap water was used for age-matched control pups (Ctrl rats; *n* = 18). The same experimental methodology was applied to all animals to offer a thorough functional and morphological assessment throughout the animal model development, at 12, 20 and 28 weeks of age, for a longitudinal evaluation. All experimental procedures were approved by the Academic Medical Center of Lisbon’s (CAML) Ethics Committee, which complied with Portuguese national and international laws governing animal care.

### 2.2. Behavioral Evaluation

As described previously elsewhere [49], animals completed a series of routine behavioral tests two weeks prior to functional evaluation to gauge their spontaneous movement and exploration employing the open-field test [50] and episodic long-term memory by novel object recognition test [51]. During the experimental days, animals were brought into the behavior testing room at least one hour before the testing session started. All behavioral experiments were carried out between 8 a.m. and 6 p.m. in a quiet room with dim lighting. All animals were handled for four days to allow them get used to the researcher and the testing environment [52]. All behavior equipment was cleaned with 70% ethanol in the intervals between animals. A UV camera (Chacon, Wavre, Belgium) was used to record all experiments, and ANY-maze software was then used to analyze the videos (Stoelting Co., Wood Dale, IL, USA).

#### 2.2.1. Open-Field Exploration Test

The open-field test (OFT) offers a distinct possibility to methodically gauge the exploratory behavior of mice in unfamiliar environments and overall locomotion activity, as well as perform a preliminary indirect screening for anxiety-related behavior [53]. This device consists of a square black box measuring 67 × 67 × 57 cm in height that is “virtually” split into three concentric squares: the periphery zone (near the walls), the middle zone, and the center. Five minutes was the average amount of time that the animals were left in the maze for parameter assessment. We determined the animals’ overall travelled distance and average speed [46,49,50,54,55].

#### 2.2.2. Novel Object Recognition Test

The novel object recognition (NOR) test was used with a 24 h retention period to assess changes in long-term memory in the animals [51]. This test occurred at the open-field test (OFT) arena. Transparent and brown glass forms that were proportional to the size of the animals were used as objects, which were randomized and used interchangeably between trials. Additionally, their placement in relation to the other objects was altered such that each object might serve as a novel or familiar [46,49,51,56].

The evaluation process involved three phases—habituation, training and testing. During the habituation phase, each animal was allowed to roam free for 15 min in the OFT arena for three consecutive days. The animal was then exposed to two sample objects (S and S’ objects) for five minutes on the fourth day, the training phase. After being exposed to the sample objects, the animal was put back in its original cage for 24 h. On the fifth day (the test phase), for five minutes the animal was exposed to two objects: a sample object (S) that had previously been explored and a novel item (N) [51]. Training and testing days were recorded and analyzed by 3-point analysis (head, torso, and tail of the animal) utilizing ANY-maze^®^ software, version 7.2. In both the training and testing stages, the duration of time the animals spent around each item was used to measure exploratory behavior. The number of approaches that involved touching, rearing toward, or sniffing the object was recorded. It was not considered exploration to approach an item from behind or cross in front without pointing the nose in its direction [51]. Exploration time was quantified as follows: ET (%) = (time exploring the object/overall exploring time) × 100.

The novelty index was calculated from NOR testing day data as follows:(ET% Novel − ET% Sample)/(ET% Novel + ET% Sample).

This index ranges from −1 to 1, with a value of 1 corresponding to the investigation of the Novelty object alone. Negative values (up to 0) show the absence of discriminating between the novel and familiar objects, i.e., spending more time examining the sample object or equally studying both objects [45,46,49,56].

### 2.3. Functional Evaluation

#### 2.3.1. Metabolic Evaluation

Before the acute surgery, at 12, 20 and 28 weeks of age, rats were housed in metabolic cages for 24 h for evaluation of food and water consumption and fecal and urine excretion.

#### 2.3.2. Acute Physiological and Autonomic Evaluation

Each animal from the experimental procedure was anesthetized at 12, 20, and 28 weeks of age with sodium pentobarbital (60 mg/kg, i.p.) and maintained, if needed, with a 20% solution (*v*/*v*) of the same anesthetic after assessing the withdrawal response. The rectal temperature was maintained between 38.5 and 39.5 °C using a homoeothermic blanket linked to a rectal probe (Harvard Apparatus, Cambourne, UK). The trachea was cannulated below the larynx to measure tracheal pressure and provide artificial ventilation. Saline and medicine injections were infused by femoral veins, while blood pressure was measured by femoral artery. Subcutaneous electrodes were used in three of the four limbs to record the electrocardiogram (ECG), and ECG data were used to determine heart rate (Neurolog, Digitimer, Welwyn Garden City, UK). Lobeline (0.2 mL, 25 μg/mL, Sigma, St. Louis, MO, USA) was injected through a catheter placed in the right carotid for pharmacological chemoreceptor stimulation [47]. Phenylephrine was injected in the femoral vein for baroreceptor reflex stimulation (0.2 mL, 25 µg/mL, Sigma, St. Louis, MO, USA) [46,47,49]. To allow for recovery to baseline levels, there were at least 3 min between each provocation. At the beginning of the experiment, an identical volume of saline was injected as a control. This showed no impact on the variables that were measured.

At the beginning of the experimental protocol, when the physiological parameters had stabilized for the subsequent autonomic analysis, a baseline recording of 10 min was obtained. Tracheal pressure, heart rate, blood pressure, ECG and respiratory rate were continuously monitored and recorded throughout the experiment (PowerLab, AD Instruments, Colorado Springs, CO, USA).

Blood from femoral artery was withdrawn at the end of the experimental protocol and atomic absorption spectrophotometer (Shimadzu, Model no. AA 7000, Kyoto, Japan) was used to evaluate the blood lead levels (BLL). The animal was then sacrificed by an anesthetic overdose, and the brain removed.

#### 2.3.3. Data Acquisition and Analysis

A frequency of 1 kHz was used to acquire, amplify, and filter all the recorded variables (Neurolog, Digitimer, Welwyn Garden City, UK; PowerLab, AD Instruments, Colorado Springs, CO, USA).

#### 2.3.4. Baro- and Chemoreceptor Reflex Evaluation

The baroreceptor reflex gain (BRG), used to assess baroreceptor reflex function, was defined by calculating the variation of HR in relation to mean BP variation during phenylephrine stimulation:ΔHR⁄ΔBP.

Basal respiratory frequency (RF, in cpm) before (on average, for 30 s) and during lobeline stimulation were used to compute the chemoreceptor response evoked by the intracarotid injection of lobeline. This formula is Δ chemoreflex (lob) = RFstimulation − RFbasal.

### 2.4. Immunohistochemistry (IHC)

The brains of 20- and 28-week animals were kept at 4 °C overnight for post-fixation in 4% paraformaldehyde (PFA) solution in phosphate buffer (pH 7.4) solution. The brains were then soaked in increasing concentrations of sucrose (15% and 30%), embedded in gelatin (7.5% gelatin in 15% sucrose solution), frozen with liquid nitrogen and 2-metilbutane (Sigma-Aldrich, Dorset, UK), and kept at 80 °C for later analysis.

To evaluate neurodegeneration, synaptic changes, astrogliosis, and microgliosis, the hippocampus (AP = −2.92 to 5.04 mm) was identified. Coronal slices (25 μm) of the hippocampus were cut using a cryostat (Leica CM 3050S, Leica Microsystems, Wetzler, Germany) and collected in a 12-well plate for storage at −20 °C in a cryoprotectant solution. The immunohistochemistry procedure was followed as previously reported [45,46,49]. Briefly, an antigen retrieval protocol [57], permeabilization with 0.3% Triton X-100 (Sigma-Aldrich, Dorset, UK), and blocking with 5% normal goat serum (BioWest, Nuaillé, France) and 1% bovine serum (VWR, Radnor, PA, USA) were performed. Tissues were immunoassayed with NeuN (1:500), Synaptophysin (1:200), GFAP (1:500), and Iba-1 (1:250) (Abcam, Cambridge, UK) primary rabbit polyclonal antibodies diluted in blocking solution overnight at 4 °C. Following three TBS washes, sections were incubated with goat anti-rabbit IgG Alexa Fluor^®^ 594 (1:1000; Thermo Fisher, Waltham, MA, USA) for tissues stained with NeuN and with goat anti-rabbit IgG Alexa Fluor^®^ 488 (1:1000; Thermo Fisher, Waltham, MA, USA) for tissues stained with Syn, GFAP, and Iba-1. The slides were mounted on Super-Frost^®^ Microscope Slides utilizing ProLong Gold Antifade with DAPI for nuclear staining (Sigma-Aldrich, Gillingham, UK). Images of the dentate gyrus region were obtained with a confocal point-scanning microscope (Zeiss LSM 880 with Airyscan, Carl Zeiss AG, Oberkochen, Germany) and post-analyzed and quantified using Fiji [58]. GFAP- and Iba-1-stained cells were morphologically classified into distinct categories of glial cells (astrocytes and microglia, correspondingly) [59,60,61,62], and positive cells for GFAP and Iba-1 were manually counted. Using the in-house program Multichannel Cell Counter RGB, the number of NeuN-positive cells (i.e., mature neurons) was computed and quantified.

### 2.5. Statistical Analysis

Unless otherwise stated, data are presented as mean ± SD and displayed as the average of the mean values across all animals. The D’Agostino and Pearson normality test was used to investigate the continuous variables’ normality distribution, and Levene’s test was applied to determine whether the variance was homogeneous. Tukey’s multiple comparisons test was applied to analyze intra- and inter-group data for the IntPb and the Ctrl groups at various time points using a two-way ANOVA (12, 20 and 28 weeks). The exploration time % between objects used in the NOR test for each group at each timepoint was also calculated using the student *t*-test for paired observations. GraphPad Prism 9 was used to analyze all the data (GraphPad Software Inc., Boston, MA, USA). Statistical significance was defined as *p* < 0.05.

## 3. Results

### 3.1. Behavioral Changes

#### 3.1.1. Intermittent Lead Exposure Does Not Cause Significant Changes in Locomotion and Anxiety-like Behavior

The open-field test was conducted to access the locomotion and exploration activity as well as anxiety-like behavior of the animals. We observed that the total number of entries in the different zones of the apparatus did not change throughout time (Figure 1a: 12 weeks—Ctrl 71 ± 18 vs. IntPb 57 ± 18; 20 weeks—Ctrl 51 ± 21 vs. IntPb 50 ± 23; 28 weeks—Ctrl 48 ± 19 vs. IntPb 52 ± 29; *p* > 0.05). In addition, there were no significant changes within groups over time.

Interestingly, we observed a significant decrease in the time spent in the center of the apparatus after the first lead exposure (Figure 1b: 12 weeks—Ctrl 5.21 ± 2.69% vs. IntPb 2.01 ± 1.04%; *p* < 0.001), a change that was not observed in the other timepoints (20 weeks: Ctrl 3.38 ± 2.66% vs. IntPb 2.09 ± 1.49%; 28 weeks: Ctrl 1.84 ± 1.15% vs. IntPb 1.35 ± 0.82%, *p* > 0.05). A significant decrease in the control group was observed over time (Ctrl 12 weeks vs. Ctrl 20 weeks, *p* < 0.05; Ctrl 12 weeks vs. Ctrl 28 weeks, *p* < 0.0001). No significant changes were observed over time in the IntPb group.

No significant intra and inter-group changes were observed in the total travelled distance throughout the time of the experiment (Figure 1c: 12 weeks—Ctrl 2006 ± 522.5 vs. IntPb 2003 ± 596.3; 20 weeks—Ctrl 1668 ± 496.8 cm vs. IntPb 1823 ± 634.6 cm; 28 weeks—Ctrl 2154 ± 837.0 cm vs. IntPb 2430 ± 986.5 cm).

We observed a significant increase in the average velocity at 12 weeks of age after the initial lead exposure (Figure 1d—Ctrl 8.72 ± 2.36 cm/s vs. IntPb 10.99 ± 4.13 cm/s, *p* < 0.05), an alteration that did not persist after a period without lead exposure (20 weeks—Ctrl 8.54 ± 2.77 cm/s vs. IntPb 9.283 ± 3.27 cm/s, *p* > 0.05) and a second lead exposure period (28 weeks—Ctrl 11.90 ± 5.72 cm/s vs. IntPb 14.32 ± 3.83 cm/s, *p* > 0.05). From the intragroup evaluation, we observed a significant increase between 20 and 28 weeks of the IntPb group (20 weeks vs. 28 weeks, *p* < 0.01).

#### 3.1.2. Animals Exposed to Lead Show Impaired Recognition of the Novel Object with a Strong Decrease in the Novel Object Recognition Index within All Evaluated Time Points

The percentage of exploration time calculated from the time the animals spent exploring the sample and the novel objects on the test day of the novel object recognition test showed that the control animals recognized the novel object as a novelty by spending more time exploring these at all time points compared to the sample object that has been previously explored on the training day (Figure 2a: 12 weeks—Ctrl *p* < 0.01, IntPb *p* > 0.05; 20 weeks—Ctrl *p* < 0.001, IntPb *p* > 0.05; 28 weeks Ctrl *p* < 0.001, IntPb *p* > 0.05).

Consequently, we observed a strong decrease in the novelty recognition index at all time points in the lead-exposed group, decreasing over time (Figure 2b: 12 weeks—Ctrl 0.2005 ± 0.2268 vs. IntPb − 0.0007 ± 0.2190; 20 weeks—Ctrl 0.1262 ± 0.1267 vs. IntPb − 0.0574 ± 0.1310; 28 weeks—Ctrl 0.2138 ± 0.1733 vs. IntPb −0.0631 ± 0.2975). No intragroup changes were observed between time points in both control and intermittent groups. Overall, these results suggest an impairment of the recognition of the objects with a possible dysfunction in the hippocampal region.

### 3.2. Physiological and Autonomic Changes

#### 3.2.1. Animals Exposed to Lead Intermittently Show a Significant Increase in Blood Lead Levels, without Effects on Metabolic Parameters

Blood lead levels were analyzed by atomic absorption spectrophotometry and the results show that IntPb groups present a strong increase in the levels at 12 weeks, decreasing after a period of non-exposure, which were still above lead levels of concern by WHO (5 µg/dL), and increasing once again to high levels after the second exposure (Table 1). We observed a significant decrease in the liquid intake at 12 weeks in the IntPb group compared to controls (*p* < 0.05). No changes were observed in the food intake and feces and urine production at all timepoints (12, 20 and 28 weeks) as well as in the liquid intake at 20 and 28 weeks. Likewise, no significant differences in weight have been observed in the animals through time.

#### 3.2.2. Lead-Exposed Animals Presented an Increase in Blood Pressure and Respiratory Frequency without Changes in Heart Rate

Cardiovascular and respiratory functions were evaluated at all timepoints in the anesthetized animals. The changes in blood pressure data are shown in Figure 3a. Animals exposed to lead showed a significant increase in systolic blood pressure at 12 and 20 weeks, after first lead exposure and post lead-free period, respectively, when compared to the control group (12 weeks—Ctrl 125 ± 26.19 mmHg vs. IntPb 164 ± 22.45 mmHg, *p* < 0.0001; 20 weeks Ctrl 137.2 ± 13.40 mmHg vs. IntPb 203.9 ± 16.32 mmHg, *p* < 0.0001), without significant changes at 28 weeks of age (28 weeks—Ctrl 120.1 ± 30.43 mmHg vs. IntPb 141 ± 23.57 mmHg, *p* > 0.05). The intragroup analysis showed an increase between 12 and 20 weeks of age and a significant decrease between 20 and 28 weeks of age in the IntPb group (12 weeks vs. 20 weeks, *p* < 0.01; 20 weeks vs. 28 weeks, *p* < 0.0001).

The changes in diastolic blood pressure are in line with those observed for systolic blood pressure. A significant increase in the diastolic blood pressure was observed in the lead-exposed group at all time points (12 weeks—Ctrl 91 ± 26.19 mmHg vs. IntPb 129 ± 14.97 mmHg, *p* < 0.0001; 20 weeks—Ctrl 104 ± 13.76 mmHg vs. IntPb 149.2 ± 10.07 mmHg, *p* < 0.0001; 28 weeks—Ctrl 90.34 ± 22.27 mmHg vs. IntPb 107.9 ± 6.013 mmHg, *p* < 0.05), as well as a significant decrease between 12 and 28 weeks of age (12 weeks vs. 28 weeks, *p* < 0.05) and 20 and 28 weeks (20 weeks vs. 28 weeks, *p* < 0.0001) in the IntPb group.

Consequently, mean blood pressure was overall significantly increased in the animals exposed to lead over time compared to the control group (12 weeks—Ctrl 113 ± 14.97 mmHg vs. IntPb 141 ± 14.97 mmHg, *p* < 0.0001; 20 weeks—Ctrl 119.4 ± 11.07 mmHg vs. IntPb 169.7 ± 12.12 mmHg, *p* < 0.0001; 28 weeks—Ctrl 104.3 ± 24.99 mmHg vs. IntPb 124.2 ± 12.59 mmHg, *p* < 0.05). Significant increase between 12 and 20 weeks (*p* < 0.01) and a decrease between 20 and 28 weeks (*p* < 0.0001) were observed when comparing different time points within the IntPb group.

No significant changes were observed inter and intragroup in heart rate data (Figure 3b: 12 weeks—Ctrl 432 ± 26.19 bpm vs. IntPb 395 ± 101.0 bpm; 20 weeks—Ctrl 433.3 ± 31.68 bpm vs. IntPb 420.9 ± 20.39 bpm; 28 weeks—Ctrl 433.5 ± 77.16 bpm vs. IntPb 404 ± 34.37 bpm, *p* > 0.05).

Regarding respiratory frequency (Figure 3c), a strong, significant increase was observed after the first lead exposure at week 12 (Ctrl 78 ± 7.48 cpm vs. IntPb 113 ± 22.45 cpm, *p* < 0.0001) with recovery after a lead-free period, at 20 weeks (Ctrl 81.65 ± 8.74 cpm vs. IntPb 81.48 ± 11.25 cpm, *p* > 0.05) and another increase at 28 weeks, after the second exposure to lead (Ctrl 60.09 ± 10.07 cpm vs. IntPb 71.85 ± 6.55 cpm, *p* < 0.05). A significant decrease between 12 and 20 weeks (*p* < 0.0001), and 12 and 28 weeks was observed in the IntPb group (*p* < 0.0001).

#### 3.2.3. Presence of Lead Caused Strong Baroreflex Impairment and Increased Chemoreceptor Reflex Sensitivity

Autonomic reflexes were pharmacologically provoked and the data are shown in Figure 4. A strong, significant decrease in baroreceptor reflex gain was observed at both 12 and 28 weeks of age after both periods of lead exposures (Figure 4a: 12 weeks—Ctrl 0.53 ± 0.37 bpm^2^/mmHg vs. IntPb 0.27 ± 0.11 bpm^2^/mmHg, *p* < 0.05; 28 weeks—Ctrl 0.77 ± 0.21 bpm^2^/mmHg vs. IntPb 0.49 ± 0.10 bpm^2^/mmHg, *p* < 0.01). Removal of lead from the diet resulted in recovery of baroreflex gain, with no significant difference observed between groups at 20 weeks (Ctrl 0.66 ± 0.31 bpm^2^/mmHg vs. IntPb 0.38 ± 0.20 bpm^2^/mmHg, *p* > 0.05). Within the intermittent lead exposure group, no significant changes were observed between the time points.

Regarding chemoreceptor reflex sensitivity, a significant increase in the data was observed at all time points in the lead-exposed group when compared to the control group (Figure 4b: 12 weeks—Ctrl 15 ± 9.73 cpm vs. IntPb 22 ± 6.74 cpm, *p* < 0.05; 20 weeks—Ctrl 20 ± 3.37 cpm vs. IntPb 31.31 ± 5.02 cpm, *p* < 0.01; 28 weeks—Ctrl13.54 ± 3.61 cpm vs. IntPb 31.85 ± 12.45 cpm, *p* < 0.001). No significant changes were detected in the intergroup analysis.

### 3.3. Molecular Changes

#### 3.3.1. Intermittent Lead Exposure Caused an Increased Number of Astrocytes and Microglia with Underlying Morphological Changes

Astrocytes were stained with GFAP marker, a filament protein mainly produced by these cells and analyzed qualitatively and quantitatively. We used pertinent papers [59,60,61] and determined that the astrocytes are activated in the lead-exposed groups as evidenced by hypertrophy of cellular processes and GFAP upregulation and higher density in the astrocytic cells, shown in the representative images (Figure 5a). We also observed an increase in the number of astrocytic cells in the intermittent group of animals according to quantitative analysis at both 20 weeks (Ctrl 128.5 ± 4.95 vs. IntPb 184.0 ± 2.00, *p* < 0.0001) and 28 weeks (Ctrl 133.7 ± 5.51 vs. IntPb 192.0 ± 5.29, *p* < 0.0001) compared to the control group. No significant differences were observed in the intragroup analysis.

Microglial cells were stained with Iba1 antibody. We determined that when the lead solution is not included in the diet, neither structural changes nor the number of cells show significant differences between IntPb and Ctrl group at 20 weeks of age (Ctrl 28.50 ± 0.71 vs. IntPb 32.67 ± 1.53, *p* > 0.05). However, once lead was reintroduced in the diet, after the second lead exposure, at 28 weeks of age, we observed a change in the microglial morphology with a loss of branching processes, an upregulation of Iba1, and a significant increase in the number of cells (Ctrl 16.00 ± 3.61 vs. IntPb 42.00 ± 5.57, *p* < 0.001). The intragroup analysis showed no significant changes in the Ctrl and IntPb groups between the different time points. These results suggest that lead induces a state of noxious microglial activation, which could be the result of a defense reaction against the neural injury caused by lead.

#### 3.3.2. Intermittent Lead Exposure Causes a Decrease in the Synaptic Marker

Synaptophysin was used as a pre-synaptic marker to assess synaptic plasticity and integrity, and a quantitative analysis was performed. We observed that the lead-exposed group, regardless of the presence of lead in the diet, displayed a significant decrease in the synaptic marker compared to the control group (Figure 6a,c—20 weeks: Ctrl 42.44 ± 11.58 vs. IntPb 23.48 ± 4.295, *p* < 0.05; 28 weeks: Ctrl 41.23 ± 8.497 vs. IntPb 18.35 ± 2.098, *p* < 0.01). This suggests that, in addition to the physiological influence of the age of the animals on the protein expression, animals exposed to lead also showed a significant marked change in neuronal dynamics at the synaptic level. No significant differences were observed in the groups between time-points; however, older animals showed a trend for a reduced expression that may express the influence of age on this synaptic marker, with lead possibly causing a higher rate of synaptic degradation.

Interestingly, quantitative analysis of the NeuN marker, which stains mature neuronal cells, showed no significant differences between the group exposed to lead and control group at either time point (Figure 6b,d—20 weeks: Ctrl 529.7 ± 32.81 vs. IntPb 486.3 ± 152.0, *p* > 0.05; 28 weeks: Ctrl 702.0 ± 212.6 vs. IntPb 540.3 ± 92.10, *p* > 0.05) and between timepoints with the intragroup analysis. Although not statistically significant, our results show a trend towards impairment of adult neurogenesis at the hippocampal level induced by lead poisoning.

## 4. Discussion

The present study aimed to understand the molecular changes involved in the temporal remodeling of physiological functions during intermittent lead exposure by drawing attention to the involvement of a limbic system structure. For that, a longitudinal study was conducted in an animal model with lead exposure focusing on the hippocampus and their projections to other central areas controlling autonomic function. Our results show activation of microglia and astrocytes in the hippocampus with concomitant local neuroinflammation, consistent with modifications in the locomotor activity of the animals, and changes in reflex regulation of cardiovascular and respiratory parameters as observed by a general increase in respiratory rate and blood pressure without changes in heart rate, a blunted baroreflex and a progressive increase in chemoreflex sensitivity. Some of these physiological changes showed a partial reversal towards normality after a period of abstinence without lead consumption.

We identified increases in GFAP and Iba1 expression in the hippocampus following intermittent lead exposure. We evaluated astrocyte- and microglial-specific cell markers to detect disruptions within the glial compartment following lead exposure.

The intermittent lead exposure (at 28 weeks of age) altered the microglia into an ameboid shape; decreased branch number and junctions; with a consequent enhancement of the Iba-1, increased the GFAP expression and increased the number of branches in the astrocytes; and decreased the synaptophysin immunoreactivity. Interestingly, we did not observe the same effect after a period of no exposure (at 20 weeks of age).

These data show that lead exposure caused microglial and astroglial activation leading to neuroinflammation. In fact, reactive astrocytes may produce pro-inflammatory cytokines, which are the primary effectors of the neuroinflammatory signals [63,64]. However, the astrogliosis observed in our model may be an indirect effect, since this astrocytic activation can occur via pro-inflammatory factors released by microglia, which becomes active and contributes to the release of more pro-inflammatory factors.

In fact, there is evidence that shows that the activity of astrocytes is critical in determining the behavioral outputs of the amygdala and hypothalamus through a process that includes the regulatory activity of specific synapses by activated astrocytes [65,66]

GFAP is a protein expressed in astrocytes in the CNS and a marker of astroglial injury [67]. One of the most recognized features of astrocytes in the mature brain is the reaction to CNS damage with reactive gliosis. The latter is characterized by the presence of large numbers of reactive astrocytes, distinguished from normal astrocytes by their greater size, longer and thicker processes, and increased GFAP levels [68]. Therefore, the increased expression of GFAP observed in intermittent Pb exposure can represent a highly sensitive marker of neurotoxin-induced CNS injury [69]. Similar phenomenon has been demonstrated by chronic lead exposure, and our previous results also demonstrated that Pb exposure since the fetal period could lead to behavioral deficits in the offspring [45,46,49,70,71]

In this study, we observed that this type of exposure produced a significant long-term episodic memory impairment (NOR test results) together with an increase in hippocampal astrocytes and microglial cells. Accumulating evidence confirms that chronic lead exposure can produce behavioral disturbances, including anxiety, in human and in animal models [72,73]. In addition, there is strong evidence suggesting an association between hippocampal dysfunction and the behavioral deficiencies observed in experimental animals following neonatal Pb exposure [43,74]. The hippocampus is critical for spatial navigation and memory. In the long-term episodic memory assessed by the novel object recognition test, we observed a strong decrease in the novelty recognition index in all time points in the lead-exposed group which becomes worse over time. It means that lead promoted cognitive dysfunction that can be in part explained by the expression of synaptophysin, astrocytes, and microglia in the dentate gyrus region, particularly affecting the recognition memory. According to our previous studies, these adverse effects were more severe in permanent or chronic exposure to lead and are reversible when the lead is withdrawn [44,45,49]

The effect of lead on cognitive parameters has been extensively reported in chronic exposure to lead, especially in memory and learning [3,8,72,75].In addition, lead exposure disturbs synaptogenesis, since the animals intermittently exposed to lead displayed a decrease in the number of synapses in the dentate gyrus region, which could lead to impairment of synaptic plasticity in the hippocampus [2,76,77]. Thus, the nervous system is the primary target for lead exposure and the developing brain appears to be especially susceptible [78]. The mechanisms of lead exposure on brain deficits remain unclear but include the microgliosis and astrogliosis observed, leading to the inflammatory cascade, which may interrupt pathways involved in the hippocampal functions [79]. Hence, these molecular changes in the brain are impactful as they may contribute to the pathophysiology of behavioral deficits and cardiovascular complications observed in intermittent lead exposure.

The autonomic reflex arc has the most integration centers in the brain, forming the central autonomic network [80,81,82,83]. There, the cardiovascular and respiratory functions are reflexively controlled to maintain the internal and external balance of the body, which is associated with the maintenance of life [84,85]. Under normal circumstances, the reflex control of the cardiovascular and respiratory systems occurs via the brainstem [83,84]. However, in situations where there is internal or external impairment (e.g., lead poisoning), central control of these functions is overtaken by the hypothalamus, which coordinates not only autonomic but also endocrine and behavioral responses to maintain homeostasis [86]. As far as cardiovascular regulation is concerned, under these circumstances, the baroreflex is inhibited, while the chemoreflex is facilitated. This alteration in the coordination of autonomic functions is observed in various pathologies or extreme physiological conditions [82,86,87].

The main cardiovascular change observed in our study and extensively reported in other lead exposure studies is hypertension. Specifically, we observed that intermittent exposure increased systolic blood pressure, diastolic blood pressure and mean arterial pressure over time with stronger effects on diastolic and mean blood pressure at the later stages. In the case of lead exposure, increased neuroinflammation and sympathetic tone can contribute to the incidence and maintenance of hypertension. Using an anti-inflammatory agent or overexpression of interleukin-10 in the brain attenuates hypertension [32,88]. In addition, the activation of microglia was observed in angiotensin II and L-NG-nitro-l-arginine methyl ester hypertension models [35]. By inhibiting microglial activation, hypertension and the neuroinflammation in the paraventricular nucleus of the hypothalamus (PVN) were attenuated together with the plasma vasopressin level and kidney norepinephrine concentration [35]. Another study using minocycline, an inhibitor of microglial activation, in spontaneously hypertensive rat (SHR) models promoted a reduction in sympathetic activity and blood pressure values [89]. Therefore, there is evidence that microglia are central to neuroinflammation and neuronal regulation of hypertension. However, the exact molecular mechanisms by which neuroinflammation regulates blood pressure remain unclear.

Besides hypertension, our results also suggest that intermittent exposure to lead decreases baroreflex sensitivity and may have occurred to counteract the increase in the blood pressure values. However, this physiological change showed a reversal towards normality after a period of abstinence (without lead consumption—at 20 weeks). These changes in baroreceptor reflex are likely mediated by changes in peripheral vascular resistance, which, in turn, are mediated by local metabolic factors (such as nitric oxide) or neurohormonal activation and are important for acute blood pressure regulation [44]. Norepinephrine released from sympathetic nerves and circulating norepinephrine released from the adrenal medulla can bind to α1 adrenergic receptors leading to vasoconstriction [90,91].

Despite the recorded hypertension and baroreflex dysfunction evoked by lead exposure, these animals did not have significant changes in heart rate when compared to controls. This leads us to conclude that intermittent lead exposure does not affect heart rate. In contrast, others have shown that developmental exposure to other toxicants, such as manganese or mercury, can increase heart rate [44,92].

Moreover, we observed that intermittent lead exposure significantly increased respiratory frequency. This increase was concomitant with a higher chemoreceptor reflex sensitivity, that, despite a lead-free period, did not show significant improvements in chemoreflex function. This indicates that lead triggered an overall alert-like reaction which could contribute not only to a higher respiratory rate but also to a blood pressure increase and the hippocampal changes observed (microglial activation and neuroinflammation). Similar results were observed in different types of lead exposure, and are similar to other conditions, such as hypertension, acute heart ischemia, or heart failure [47,81,93,94].

Regarding blood lead level (BLL), we determined that the concentration at 20 weeks of age is very low compared to the BLL at 12 and 28 weeks of age. This is because these animals had no exposure (tap water) until 20 weeks of age and lead is degraded every 40 days. This time period was chosen to show that although lead is excreted from the blood and has a low BLL, it still has various adverse health effects on the body. These data are also concomitant with the changes that are happening in the systematic level, the strongest occurring at 28 weeks, after the second exposure. We understand that the present work has some limitations, such as the lack of evaluation of lead in soft tissues and bones, which are the primary Pb storages; however, as blood lead levels are the main biomarker for lead exposure, we have chosen this particular parameter to maximize the translation to human environmental lead exposure and its detection.

Therefore, our data indicate that lead exposure from the fetal period induces a permanent chemoreceptor dysfunction and a baroreceptor impairment that can be partially responsible for high blood pressure values. This dysfunction could have been evoked by fetal exposure to lead, resulting in impairment of nervous system development.

## 5. Conclusions

In conclusion, the present study demonstrates a change in the crosstalk between astrocytes and microglia associated to neuroinflammation at hippocampal level during intermittent lead exposure. This chronic glial activation was accompanied by changes in behavioral and the central reflex regulation of cardiorespiratory functions, some of them with a trend to a partial reversal to physiological values after a blanking period without lead exposure showing the ability of the autonomic nervous system to remodel reversely after injury and under the appropriate environment. In addition, our results suggest that chronic neuroinflammation promoted by lead exposure may increase the susceptibility to adverse events in individuals with pre-existing cardiovascular disease and/or in the elderly. Our longitudinal study is the first to our knowledge that shows the evolution of low-level intermittent lead exposure from the fetal period until adulthood, which could contribute to development and implementation of the prevention and public policy strategies.

## Figures and Tables

**Figure 1 cells-12-00818-f001:**
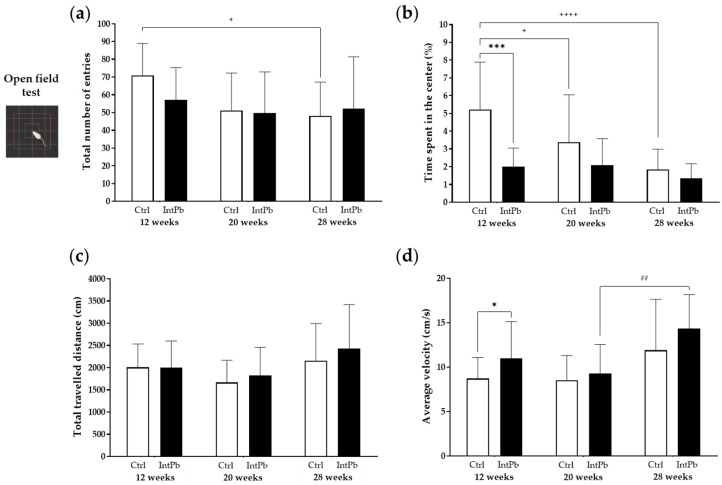
Locomotor and exploratory behaviors and anxiety-like behavior assessed by the open-field test: (**a**) Total number of entries in the different regions. (**b**) Time spent in the central zone. (**c**) Total travelled distance in the apparatus. (**d**) Average velocity of the animals in the apparatus. Values are expressed as the mean ± SD for *n*_IntPb_ = 15 and *n*_Ctrl_ = 18 for each evaluated time-point. Symbols denote statistically significant differences inter (Ctrl vs. IntPb—* *p* < 0.05; *** *p* < 0.001) and intra groups (Ctrl 12 weeks vs. Ctrl 20 weeks vs. 28 weeks—+ *p* < 0.05, ++++ *p* < 0.0001; IntPb 12 weeks vs. IntPb 20 weeks vs. 28 weeks—## *p* < 0.01,); two-way ANOVA, Tuckey’s multiple comparison test.

**Figure 2 cells-12-00818-f002:**
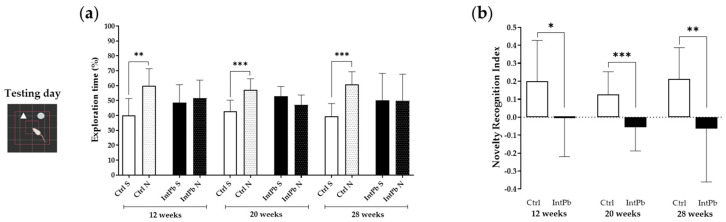
Episodic long-term memory assessed by novel object recognition test. (**a**) Percentage of exploration time of sample (S) and novel (N) objects by each group at the different time points. (**b**) Novelty recognition index is calculated by the equation presented in the Section 2. Values are expressed as the mean ± SD for *n*_IntPb_ = 15 and *n*_Ctrl_ = 18 for each evaluated time-point. Symbols denote statistically significant differences inter groups; * *p* < 0.05 ** *p* < 0.01, *** *p* < 0.001; paired Student’s *t*-test (**a**) and two-way ANOVA, Tukey’s multiple comparison test (**b**).

**Figure 3 cells-12-00818-f003:**
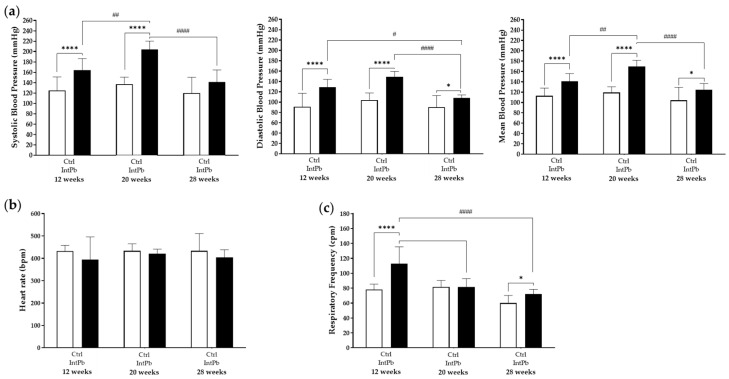
Cardiovascular and respiratory parameters evaluated along the three timepoints. (**a**) Systolic, diastolic, and mean blood pressure assessed from the femoral artery. (**b**) Heart rate values were calculated from the electrocardiogram. (**c**) Respiratory rate calculated from basal tracheal pressure. Values are expressed as the mean ± SD for *n*_IntPb_ = 12 and *n*_Ctrl_ = 12 for each evaluated time-point. The symbols denote statistically significant differences inter (Ctrl vs. IntPb—* *p* < 0.05; **** *p* < 0.001) and intra groups (IntPb 12 weeks vs. IntPb 20 weeks vs. 28 weeks—# *p* < 0.05, ## *p* < 0.01, #### *p* < 0.0001); two-way ANOVA, Tuckey’s multiple comparison test.

**Figure 4 cells-12-00818-f004:**
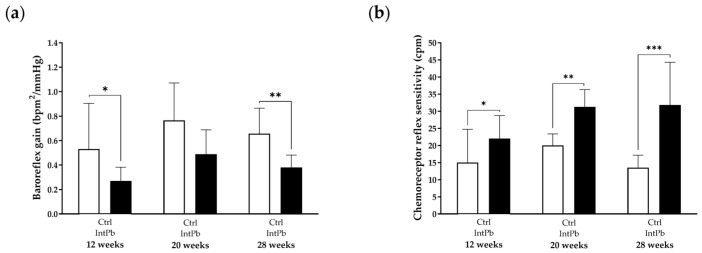
Variation of baro- and chemoreflex expression along the three time points for all groups. (**a**) Baroreflex gain calculated from the variation in heart rate and blood pressure after phenylephrine stimulation. (**b**) Chemoreceptor reflex sensitivity calculated from the variation in respiratory rate upon lobeline stimulation. Values are expressed as the mean ± SD for *n*_IntPb_ = 12 and *n*_Ctrl_ = 12 for each evaluated time-point. The symbols denote statistically significant differences inter groups (Ctrl vs. IntPb—* *p* < 0.05; ** *p* < 0.01; *** *p* < 0.001); two-way ANOVA, Tuckey’s multiple comparison test.

**Figure 5 cells-12-00818-f005:**
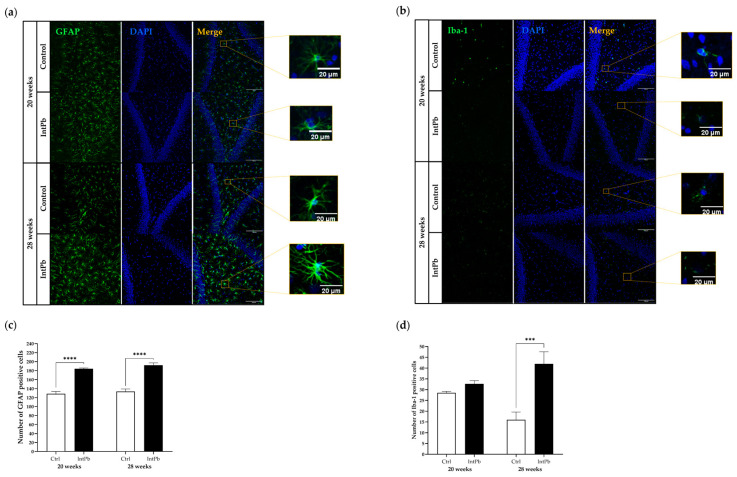
Neuroinflammation evaluated through astrocytic (GFAP) and microglial (Iba1) markers by immunohistochemistry. (**a**) Representative images of the GFAP (1:500)-stained astrocytes. (**b**) Quantification of GFAP-positive cells. (**c**) Representative images of the Iba1 (1:250)-stained microglia. (**d**) Quantification of Iba1-positive cells. Images were acquired using a confocal point scanning microscope (Zeiss LSM 880 with Airyscan) with a 20× objective. The scale bar is 50 µm or 20 µm for stained images. Values are expressed as the mean ± SD for *n*_IntPb_ = 3–4 and *n*_Ctrl_ = 3–4 for each evaluated time-point. The symbols denote statistically significant differences inter groups (Ctrl vs. IntPb—*** *p* < 0.001, **** *p* < 0.0001); two-way ANOVA, Tuckey’s multiple comparison test.

**Figure 6 cells-12-00818-f006:**
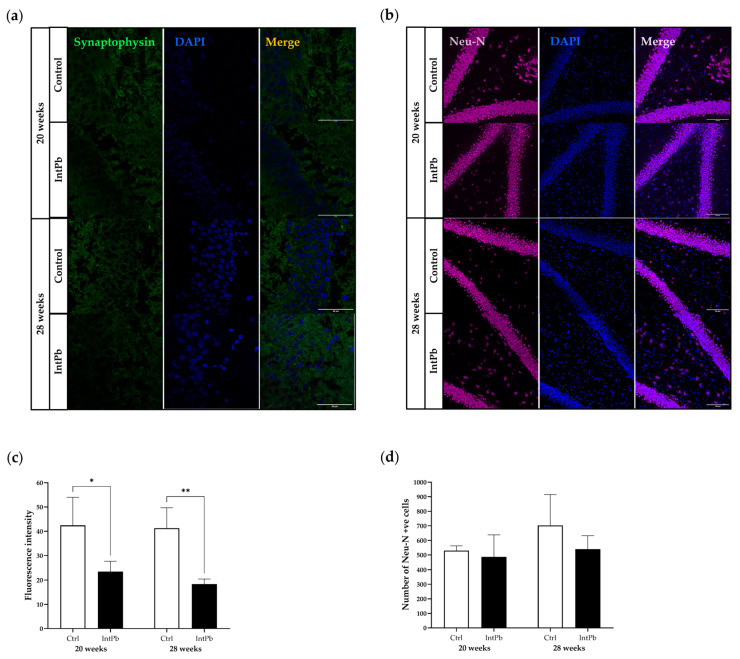
Synaptic alterations (Syn) and neurodegeneration (NeuN) results evaluated by the immunohistochemistry technique. (**a**) Representative images of the Syn (1:500)-stained pre-synapses. (**b**) Histogram of the fluorescence intensity of Syn staining. (**c**) Representative images of the NeuN (1:500)-stained neurons. (**d**) Histogram of NeuN-positive cells’ quantification. Images were acquired using a confocal point scanning microscope (Zeiss LSM 880 with Airyscan) with 20× objective. Scale bar is 50 µm for stained images. Values are expressed as the mean ± SD for *n*_IntPb_ = 3–4 and *n*_Ctrl_ = 3–4 for each evaluated time-point. The symbols denote statistically significant differences inter groups (Ctrl vs. IntPb—* *p* < 0.05, ** *p* < 0.01); two-way ANOVA, Tuckey’s multiple comparison test.

**Table 1 cells-12-00818-t001:** Blood lead levels and metabolic parameters of animals exposed to lead compared to controls. Values are the mean ± SD. The asterisks denote statistically significant differences between groups; ^ns^—not significant, * *p* < 0.05; two-way ANOVA, with multiple comparisons (Tukey’s test); *n* = 6/group.

Age	Group	Blood Lead Levels (μg/dL)	Weight (g)	Food Intake (g)	Liquid Intake (mL)	Produced Feces (g)	Produced Urine (mL)
12 weeks	Ctrl	<0.1	358 ± 95	25 ± 2	33 ± 7	12 ± 4	19 ± 4
	IntPb	24.0 ± 3.1	345 ± 92 ^ns^	22 ± 3 ^ns^	22 ± 2 *	9 ± 3 ^ns^	15 ± 4 ^ns^
20 weeks	Ctrl	<0.1	386 ± 116	23 ± 9	39 ± 8	12 ± 3	17 ± 6
	IntPb	5.8 ± 0.7	390 ± 99 ^ns^	25 ± 3 ^ns^	35 ± 0 ^ns^	8 ± 2 ^ns^	14 ± 4 ^ns^
28 weeks	Ctrl	<0.1	333 ± 101	23 ± 5	24 ± 1	8 ± 1	11 ± 3
	IntPb	20.5 ± 2.7	428 ± 123 ^ns^	24± 4 ^ns^	25 ± 5 ^ns^	12 ± 5 ^ns^	11 ± 1 ^ns^

## Data Availability

The data presented in this study are available on request from the corresponding author. The data are not publicly available due to ethical restrictions.

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
