# Peer review of "Intermittent Lead Exposure Induces Behavioral and Cardiovascular Alterations Associated with Neuroinflammation"

_cells, 2023, doi:10.3390/cells12050818_

Round 1

Reviewer 1 Report

1.     Use chemical formula of lead (Pb).

2.     Introduction: Line 86: replace in with is 

3.     Introduction: Line 89 Add reference after activation.

4.      The author should include the Pb concentration measured in the blood as well as in the brain by ICP-MS

5.     The authors have only measured only protein levels and did not paid any attention to the gene expression.

6.     Many cytokines are hard to be detected at protein level (western blotting or ELISA) but gene expression can be easily measured by qPCR.

7.     Earlier studies have pointed out epigenetic pathway that can explain the onset of neurodegeneration, the authors should explore that angle in order to give a mechanistic approach to the study.

Author Response

Comments and Suggestions for Authors

  1. Use chemical formula of lead (Pb).
    Response: Thank you for your remark. We have revised the article and changed lead to Pb in some places to create variability and easier readability.
  2. Introduction: Line 86: replace in with is 

Response: Thank you for your comment. We have checked the sentence, and we see that changing the wording will lead to poor understandinhg of this sentence. Hence, we left the same word.

  1. Introduction: Line 89 Add reference after activation.
    Response: Thank you for the comment. We have added a reference to the sentence.

  1. The author should include the Pb concentration measured in the blood as well as in the brain by ICP-MS
    Response: Thank you for your remark. We have added the blood lead levels and the general metabolic data of the animals in the methods and results sections. Unfortunately, levels of lead in the brain were not evaulated in these animals.

  1. The authors have only measured only protein levels and did not paid any attention to the gene expression.

Response: We really appreciate your remark. We see the value of the gene expression analysis, however, for a first analysis we have decided to only perform protein level evaluation. We will consider this analysis for a future project.

  1. Many cytokines are hard to be detected at protein level (western blotting or ELISA) but gene expression can be easily measured by qPCR.

Response: Thank you for your comment. We will take this remark for our future project.

  1. Earlier studies have pointed out epigenetic pathway that can explain the onset of neurodegeneration, the authors should explore that angle in order to give a mechanistic approach to the study.

Response: Thank you for the remark. Indeed, epigenetic changes have been studied and shown to be a major mecanistic change in lead exposure. However, in the present physiological and translational study, we opted to not evaluate this aspect and focused on the effects of exposure in the animal physiology. 

Reviewer 2 Report

The paper presents high quality of experimental methods linking to Lead exposure and some neuro-modulations.

In the environmental perspective, this metal is absorbed by oral via, normally.

At the occupational perspective, pulmonar via.

Is that acetate lead exposure dose was calculated/extrapolated to human ?

It was possible to quantify the blood lead level ?

The paper presents sufficient information for publication.   Only the extrapolation of animal-human aspects could be improved.

Author Response

Comments and Suggestions for Authors

The paper presents high quality of experimental methods linking to Lead exposure and some neuromodulations.

Thank you for your comment. We appreciate your consideration of our work.

In the environmental perspective, this metal is absorbed by oral via, normally.

At the occupational perspective, pulmonar via.

Is that acetate lead exposure dose was calculated/extrapolated to human ?

Response: Thank you for your remark. The dose of lead acetate that was used in this study has been extrapolated to humans and 0.2% p/v of lead acetate solution is in fact 2000ppm or 2000mg/L of the solution which is considered a low-level environmental lead exposure in humans.

It was possible to quantify the blood lead level ?

Response: Thank you for your remark. We have added the blood lead levels and the general metabolic data of the animals in the methods and results sections..

The paper presents sufficient information for publication.   Only the extrapolation of animal-human aspects could be improved.

Response: Thank you for your comment. The intermittent low-level environmental lead exposure was chosen to mimic the exposure that happens in human population in the most recent years in which immigration, globalization and exchange programs have been increased. This led to people who were born in a region with lead exposure move to study/work in a more developed region without lead in the environment. However, usually due to the drive to go back to the homeland after the prime years of studies and work, people move back to their hometowns and get exposed again.

Reviewer 3 Report

This manuscript by Shvachiy et al. provides data for the intermittent lead exposure on biochemical variables suggestive of some neurological and cardiovascular variables. Number of parameters were evaluated to support the objectives and conclusion. Authors conclude that intermittent exposure to lead induces microgliosis and astrogliosis in the rat hippocampus. In my opinion there is very little novelty in the study as lead exposure and in particular its neurological and cardiovascular effects are well known for years including mechanism of its toxicity and the current study provides very little which is new. I have few other major concerns which re listed below - 

1. Authors were not able to highlight what is the novelty in this study except the mode of exposure was chosen as intermittent ingestion. 

2. Authors also failed to convincingly justify the rationale (objective) of the study. Why the intermittent lead exposure was chosen with respect to actual human exposure? Why only a single dose was chosen and what was the basis of choosing 0.2% (p/v) lead acetate in drinking solution? 

3. No data was provided for the water , food intake or body weight? These are important variables. How much lead actually was ingested by the animals ?

4. Where is the data for the lead concentration in blood, and brain and other soft organs? These are major miss 

5. Authors also were not able to convincingly justify the selection of cardiovascular as their end point goal when lead is known to have major adverse effects on haematopoietic, renal and CNS ? They should have evaluated some of the specific lead sensitive variables to support their data or would have derived a correlation between their data and these variables. 

6. Though number of neurobehavioral variables were evaluated, it would have been ideal if some neurotransmitters could also have been evaluated. This could have given some interesting conclusion. 

7. Discussion is weak and authors failed to discuss their own results with what has already been reported. In my opinion their literature survey is inadequate as few important papers are missing which could have been discussed. 

8. Presentation too is weak as it is difficult to follow text at few places. 

Author Response

Comments and Suggestions for Authors

This manuscript by Shvachiy et al. provides data for the intermittent lead exposure on biochemical variables suggestive of some neurological and cardiovascular variables. Number of parameters were evaluated to support the objectives and conclusion. Authors conclude that intermittent exposure to lead induces microgliosis and astrogliosis in the rat hippocampus. In my opinion there is very little novelty in the study as lead exposure and in particular its neurological and cardiovascular effects are well known for years including mechanism of its toxicity and the current study provides very little which is new. I have few other major concerns which re listed below – 

Thank you for comment.

  1. Authors were not able to highlight what is the novelty in this study except the mode of exposure was chosen as intermittent ingestion. 
    Response: Thank you for the remark. We have amended the text and showed the novelty of our study in a clearer way.
  2. Authors also failed to convincingly justify the rationale (objective) of the study. Why the intermittent lead exposure was chosen with respect to actual human exposure? Why only a single dose was chosen and what was the basis of choosing 0.2% (p/v) lead acetate in drinking solution? 
    Response: We appreciate your comment. The intermittent low-level environmental lead exposure was chosen to mimic the exposure that happens in human population in the most recent years in which immigration, globalization and exchange programs have been increased. This led to people who were born in a region with lead exposure move to study/work in a more developed region without lead in the environment. However, usually due to the drive to go back to the homeland after the prime years of studies and work, people move back to their hometowns and get exposed again.

The dose of lead acetate that was used in this study has been extrapolated to humans and 0.2% p/v of lead acetate solution is in fact 2000ppm or 2000mg/L of the solution which is considered a low-level environmental lead exposure in humans.

  1. No data was provided for the water , food intake or body weight? These are important variables. How much lead actually was ingested by the animals?
    Response: Thank you for your comment. We agree with the statement and added the methodology and data of metabolic variables in all timepoints, namely water and food intake and urine and faeces production.
  2. Where is the data for the lead concentration in blood, and brain and other soft organs? These are major miss 
    Response: Thank you for your remark. We also see the need to show the blood lead levels and have included this data in the paper. We have decided to firstly do the analysis of blood lead levels and the analysis of other tissues will be done in the future.
  3. Authors also were not able to convincingly justify the selection of cardiovascular as their end point goal when lead is known to have major adverse effects on haematopoietic, renal and CNS? They should have evaluated some of the specific lead sensitive variables to support their data or would have derived a correlation between their data and these variables. 
    Response: Thank you for comments. We have chosen cardiovascular, autonomic and cognitive parameters as these, together with neurological, renal and hematopoietic changes, constitute the main health adverse effects of chronic environmental lead exposure. In fact, the hypothalamus is known to be crucially involved in the regulation of cardiovascular functions as a central link in the control of both autonomic outflow and of the hypothalamic–pituitary–adrenal (HPA) axis. We have evaluated the blood pressure to evaluate hypertension, heart rate and respiratory frequency (to check for tachypnoea) and supported the data with cardiorespiratory reflexes evaluation (baro- and chemoreceptor reflexes). This data showed that despite a lead-free period, this type of exposure induces a permanent chemoreceptor dysfunction and a baroreceptor impairment that triggered an overall alert-like reaction which could contribute not only to a higher respiratory rate but also to hypertension and to the hypothalamic changes observed (microglial activation and neuroinflammation).
  4. Though number of neurobehavioral variables were evaluated, it would have been ideal if some neurotransmitters could also have been evaluated. This could have given some interesting conclusion. 
    Response: We really appreciate your comment and agree that this type of analysis could have been interesting. However, due to the type of study and the different variables we have already evaluated, we have postponed the evaluation of neurotransmitters, focusing mainly on neuroinflammation and synaptic function. We will take this idea in consideration for a future study.
  5. Discussion is weak and authors failed to discuss their own results with what has already been reported. In my opinion their literature survey is inadequate as few important papers are missing which could have been discussed. 
    Response: Thank you for your comment. We have reviewed the discussion and introduction and made some changes for a cleared presentation. Also, we have reviewed the articles that were included in this study, and included some more recent ones.
  6. Presentation too is weak as it is difficult to follow text at few places. 
    Response: Thank you for your remark. We have reviewed the whole paper once again to make it more clearer for the readers, specially focusing in the introduction and discussion.

Round 2

Reviewer 3 Report

Authors have made significant changes in their revised manuscript however, I regret that it still has not reached the required standard for acceptance. Few critical points still remained unanswered and require clarifications.  

1. Authors failed to justify the objectives particularly the choice of intermittent exposure. It needs to be supported by some literature. It is well known that lead gets deposited in hard tissue on chronic exposure and even if a person has moved out of place of exposure, this deposited lead becomes future source of lead exposure. Why I am indicating this is for the fact that blood lead level almost reached normal level at second end point . This is unbelievable and authors need to check their data carefully. Blood lead concentration has not been discussed at all in the Discussion section. 

2. Further, authors though have included blood lead level but why other soft tissues and particularly data for hard tissue lead concentration still remained missing. This is important data and without this the study has little or no meaning 

Author Response

Authors have made significant changes in their revised manuscript however, I regret that it still has not reached the required standard for acceptance. Few critical points still remained unanswered and require clarifications.  

  1. Authors failed to justify the objectives particularly the choice of intermittent exposure. It needs to be supported by some literature. It is well known that lead gets deposited in hard tissue on chronic exposure and even if a person has moved out of place of exposure, this deposited lead becomes future source of lead exposure. Why I am indicating this is for the fact that blood lead level almost reached normal level at second end point . This is unbelievable and authors need to check their data carefully. Blood lead concentration has not been discussed at all in the Discussion section. 

Response: Thank you for your comments. We see the point of the reviewer and we have included some papers that have been elaborated previously to support our aim of the work. Lead is deposited in the soft tissues, like heart, liver and kidneys and its long-term storage is in bones, however one on the main biomarkers for the presence of lead in the body is the blood lead level (BLL) which we have measured in all timepoints. The BLL is very low at 20 weeks of age due to lead clearance in the blood which happens every 40 days and the lead exposure pause in our study is for 8 weeks which is 56 days. This period was chosen with the purpose to show how, even though, lead was cleared from the diet and has a low concentration in the blood, still causes several effects in the body. The authors saw the mistake of not presenting the blood levels data in the discussion and have added this information in the discussion sector.

  1. Further, authors though have included blood lead level but why other soft tissues and particularly data for hard tissue lead concentration still remained missing. This is important data and without this the study has little or no meaning 

Response: Thank you for your remark. We understand the issue that the reviewers is rising and see the lack of not evaluating the presence of lead in the tissue and will consider this in the next studies. We chose blood lead levels because this is the most important biomarker of lead exposure in humans, and we felt it was necessary for it to be most similar to human outcomes in a translational study.
